# 15-*cis*-Phytoene Desaturase and 15-*cis*-Phytoene Synthase Can Catalyze the Synthesis of β-Carotene and Influence the Color of Apricot Pulp

**DOI:** 10.3390/foods13020300

**Published:** 2024-01-17

**Authors:** Ningning Gou, Xuchun Zhu, Mingyu Yin, Han Zhao, Haikun Bai, Nan Jiang, Wanyu Xu, Chu Wang, Yujing Zhang, Tana Wuyun

**Affiliations:** 1Kernel-Apricot Engineering and Technology Research Center of State Forestry and Grassland Administration, Research Institute of Non-Timber Forestry, Chinese Academy of Forestry, State Key Laboratory of Tree Genetics and Breeding, Zhengzhou 450003, China; lemonn@caf.ac.cn (N.G.); ymy920916@163.com (M.Y.); zhaohan@caf.ac.cn (H.Z.); bhk1994@163.com (H.B.); south.j@outlook.com (N.J.); xwywanyu@163.com (W.X.); wangchu1226@163.com (C.W.); zhangyujing@caf.ac.cn (Y.Z.); 2School of Food and Health, Beijing Technology and Business University, Beijing 100037, China; xxcy08@163.com; 3Institute of Ecological Conservation and Restoration, Chinese Academy of Forestry, Key Laboratory of Desert Ecosystem and Global Change, National Forestry and Grassland Administration, Beijing 100091, China

**Keywords:** apricot, 15-*cis*-phytoene synthase, 15-*cis*-phytoene desaturase, β-carotene, color

## Abstract

Fruit color affects its commercial value. β-carotene is the pigment that provides color for many fruits and vegetables. However, the molecular mechanism of β-carotene metabolism during apricot ripening is largely unknown. Here, we investigated whether β-carotene content affects apricot fruit color. First, the differences in β-carotene content between orange apricot ‘JTY’ and white apricot ‘X15’ during nine developmental stages (S1–S9) were compared. β-carotene contents highly significantly differed between ‘JTY’ and ‘X15’ from S5 (color transition stage) onwards. Whole-transcriptome analysis showed that the β-carotene synthesis genes 15-*cis*-phytoene desaturase (PaPDS) and 15-*cis*-phytoene synthase (PaPSY) significantly differed between the two cultivars during the color transition stage. There was a 5 bp deletion in exon 11 of *PaPDS* in ‘X15’, which led to early termination of amino acid translation. Gene overexpression and virus-induced silencing analysis showed that truncated *PaPDS* disrupted the β-carotene biosynthesis pathway in apricot pulp, resulting in decreased β-carotene content and a white phenotype. Furthermore, virus-induced silencing analysis showed that *PaPSY* was also a key gene in β-carotene biosynthesis. These findings provide new insights into the molecular regulation of apricot carotenoids and provide a theoretical reference for breeding new cultivars of apricot.

## 1. Introduction

Color is a key indicator for evaluating fruit maturity and quality. Previous studies have shown that chlorophyll, carotenoids, anthocyanins, and betalains were the main pigments that affect fruit coloring. Carotenoids are one of the main pigments for coloration, showing yellow, orange, red, or others in plant tissue; they are important components for photosynthesis and are involved in light harvesting and photoprotection, plant growth, and development [1]. Recent studies have shown that kiwifruit [2,3], kumquat [4], tomato [5,6,7], apple [8], carrot [9], banana [10], and apricot [11] contain major accumulations of carotenoids. Furthermore, anthocyanins mainly accumulate in mulberry [12], Chinese Jujube [13], Chokecherry [14], peach [15], citrus [16], grape [17], persimmon [18], black chokeberry [19], eggplant [20], and onion [21].

Apricot (*Prunus armeniaca* L.) is a very popular fruit because of its unique flavor and bright color. However, during its production, its coloring performance can be unstable, resulting in decreased profits. Apricot, one of the most important fruit crops, has different cultivars with distinct colorations, which mainly result from variations in the compositions of carotenoids, especially β-carotene [11,22,23]. Carotenoids are the general name of a class of natural pigments, which widely exist in nature. According to the most recent compilation (2020), approximately 1204 natural carotenoids have been reported in 722 organisms [24]. Conjugated double-bond structures are prevalent in carotenoid molecules and most carotenoids appear orange, yellow, or red in visible light, except for a small number of colorless carotenoids such as phytoene and phytofluene [25,26]. Carotenoids contribute to fruit color and nutritional value, which are important for consumer preferences [27,28]. The carotenoid metabolic pathway in higher plants has been clearly defined and contains many essential enzymes [29].

To date, more than 150 genes encoding carotenoid synthase have been successfully cloned from a variety of plants and algae [30]. The first coding gene associated with carotenoid synthesis isolated from plants was *PSY*, which was considered a key speed-limiting enzyme in the carotenoid synthesis pathway. There was a strong correlation between the metabolic flux of the carotenoid synthesis pathway and *PSY* gene expression and activity [31]. Many studies have shown that overexpression of the *PSY* gene in some plants could significantly increase the content of total carotenoids and β-carotene in plants [32,33]. Colorless phytoene is continuously dehydrogenated by *PDS*, zeta-carotene isomerase (ZISO), and zeta-carotene desaturase (ZDS) to produce 7,9,7′,9′-tetra-*cis*-lycopene. At present, the *PDS* gene has been obtained in corn, sweet potato, tomato, gentiana lutea, crocus, daffodil, Dunhalina, and bacteria [34,35]. *PDS* and *ZDS* genes have been identified as single copies in tomatoes, grapes, and Arabidopsis [36,37]. The synthesis of lycopene is catalyzed by prolycopene isomerase (CRTISO) and mutations in the *CRTISO* gene in Arabidopsis, tomato [38], and melon [39] led to a large accumulation of *cis*-lycopene in seedlings and pulp. Lycopene is catalyzed by lycopene beta-cyclase (LCYB) to produce β-carotene. An insertional mutation in the *LCYβ2* gene was found to be responsible for the reddish color of papaya pulp, inactivating lycopene cyclase and accumulating large amounts of lycopene [40]. The level of β-carotene in melon was positively correlated with the expression level of *CmLCYβ1*, increasing the expression level of the *LCYβ1* gene may be the main way to increase β-carotene content [41]. The content of β-carotene was significantly decreased by overexpression of the β-carotene hydroxylase gene *AcBCH1/2* in yellow-pulp kiwifruit [3]. *FcrNAC22* mediates red-light-induced fruit coloration through up-modulated carotenoid metabolism in kumquat [4]. The transcription of the β-carotene hydroxylase gene *SlBCH1* involved in β-carotenoid catabolism was suppressed in tomato *SlIDI1* mutant fruit [7].

For the study of genes related to carotenoid metabolism in apricot fruit, researchers have focused on the expression of β-carotene degrading genes and there were few reports on the expression and functional verification of synthetic genes. Through transcriptome analysis, Jiang et al. found that the reason for the low β-carotene content in white apricot cultivars was the high expression of the 9-*cis*-epoxycarotenoid dioxygenase (NCED) gene at maturity [42]. Xi et al. found that (+)-abscisic acid 8′-hydroxylase (CYP707A), zeaxanthin epoxidase (ZEP), carotenoid 9,10(9′,10′)-cleavage dioxygenase 1 (CCD1), and carotenoid cleavage dioxygenase4 (CCD4) may be key regulatory genes for carotenoid and aromatic volatile carotenoid accumulation in apricot fruit [43]. Garcia-Gomez et al. found that *CCD4* was the key gene that caused the large difference in total carotenoid content [44]. Zhou et al. found that *NCED*1 and *CCD4* were the key genes leading to the great differences in the total carotenoid content [23].

In our earlier study, we observed significant differences in β-carotene content among apricot cultivars with different pulp colors, especially those with pure white and orange pulp apricots. However, the specific reasons for the significant differences in β-carotene content between different cultivars are not fully understood. In this study, two apricot cultivars with significant differences in β-carotene content were selected and their β-carotene content was measured at different developmental stages (S1–S9). Based on transcriptomic data, the expression of genes related to carotenoid metabolism was analyzed and the difference in pulp color between two apricot cultivars was pointed to the synthetic β-carotene genes *PaPSY* and *PaPDS*. The functions of the β-carotene synthesis structural genes *PaPSY* and *PaPDS* were verified through virus-induced gene silencing and overexpression injection in apricot fruit. Functional analysis revealed that *PaPSY* and *PaPDS* could affect the accumulation of β-carotene in apricot pulp. We also systemically identified and characterized lncRNAs and mRNAs from two apricot varieties, aiming to analyze the major mRNAs and non-coding RNAs related to pulp color differences. To date, the regulation of lncRNA in relation to apricot pulp color has not been studied. Our findings provide new insight into the molecular mechanisms underlying fruit color formation.

## 2. Materials and Methods

### 2.1. Plant Materials

Two apricot cultivars were used in this research: the white pulp apricot ‘X15’ and the orange pulp apricot ‘JTY’. The apricot cultivars were grown in the experimental field of the Research Institute of Non-timber Forestry in Yuanyang County, Henan Province, China. Apricot pulps were collected from nine stages (sampling began from the initial stage of apricot pulp cell expansion) and three trees as individual biological replicates: 49 days after full bloom (DAFB; S1); 56 DAFB (S2); 63 DAFB (S3); 70 DAFB (S4); ‘X15’ 77 DAFB; ‘JTY’, 74 DAFB (S5); ‘X15’, 84 DAFB; ‘JTY’, 77 DAFB (S6); ‘X15’, 88 DAFB; ‘JTY’, 81 DAFB (S7); ‘X15’, 91 DAFB; ‘JTY’, 84 DAFB (S8); ‘X15’, 94 DAFB; and ‘JTY’, 88 DAFB (S9). The apricot pulps used for determining β-carotene, chlorophyll a, and chlorophyll b content were collected during S1–S9. Whole-transcriptome sequencing was performed for S3 (green fruit stage), S5 (color transition stage), and S8 (commercial maturity stage), according to the changes in β-carotene content. The collected samples were quickly frozen in liquid nitrogen and stored at −80 °C until physiological index determination and RNA extraction. The sample information of different cultivars of apricots is shown in Appendix A.

### 2.2. β-Carotene and Chlorophyll Determination

Apricot pulp samples stored at −80 °C were used for grinding to obtain apricot pulp powder. In total, 50 mg pulp powder was weighed and 0.5 mL of a mixed solution of n-hexane:acetone:ethanol (1:1:1, *v*/*v*/*v*) was added. The supernatants were collected by centrifugation at 12,000 r/min for 5 min at 4 °C. Then, saturated sodium chloride solution (0.5 mL) was added to the supernatants and vortexed. This solution was evaporated to dryness and dissolved in 0.5 mL of MTBE, to which 0.5 mL of 10% KOH-MeOH was added. After the reaction, 1 mL of saturated sodium chloride solution and 0.5 mL of MTBE were added and vortexed, the upper layer was collected, and the supernatant was evaporated to dryness and reconstituted in 100 μL of MeOH/MTBE (1:1, *v*/*v*) [45]. The solution was filtered through a 0.22 μm membrane filter for further LC-MS/MS analysis. The sample extracts were analyzed using an UPLC-APCI-MS/MS system (UPLC, ExionLC™ AD, https://sciex.com.cn/ (accessed on 2 August 2021); MS, Applied Biosystems 6500 Triple Quadrupole, https://sciex.com.cn/ (accessed on 2 August 2021)). The analytical conditions were as follows: LC column, YMC C30 (3 μm, 100 mm × 2.0 mm ID); solvent system, methanol: acetonitrile (1:3, *v*/*v*) with 0.01% BHT and 0.1% formic acid (A) and methyl tert-butyl ether with 0.01% BHT (B); gradient program, started at 0% B (0–3 min), increased to 70% B (3–5 min), then increased to 95% B (5–9 min), and finally ramped back to 0% B (10–11 min); flow rate, 0.8 mL/min; temperature, 28 °C; and injection volume, 2 μL [46,47].

### 2.3. LncRNA and mRNA Deep Sequencing

Whole-transcriptome libraries were constructed and deep sequencing was performed by the Beijing Genomics Institute (Shenzhen, China). The ribosome chain specific model was used to construct the library and the method was as follows: the ribosomal RNA was removed from the total RNA, purified, recovered, and segmented under a certain temperature and ion environment. A single strand cDNA was synthesized using the fragment RNA as a template and then dUTP was used instead of dTTP to synthesize a double strand of cDNA. The double-stranded cDNA was end-repaired, a base “A” was added to the 3′ end, and a splice was attached. The double-stranded cDNA containing U-strand was digested by UDG enzyme and amplified by PCR. The quality of the constructed library was tested and the qualified library products were cyclized. The circular DNA molecules were replicated through the rolling ring to form DNA nanospheres (DNBS), which were sequenced on the DNBSEQ platform and 10 G of clean data were obtained for each sample. The identification, quantification, and functional analysis of lncRNAs and mRNAs were achieved.

### 2.4. Identification of lncRNAs and mRNAs

The short read comparison tool SOAP (version: v1.5.2) [48] was used to compare reads to the ribosome database for data filtering. Clean reads were aligned to the apricot reference genome (Genome Database for Rosaceae, tfGDR1049) using the alignment software HISAT (version: v2.0.4) [49], followed by alignment and assembly with the strain-specific patterns of HISAT [49] and StringTie (version: v1.0.4) [50]. Cuffcompare (version: v2.2.1) [51] was used to compare the assembled transcripts with known mRNAs and lncRNAs to obtain information about their mutual position relationship. According to the classification and statistics of lncRNAs in the NONCODE [52] database, five types of lncRNAs containing the most were retained, namely i (a transfrag falling entirely within a reference intron), j (at least one splice junction is shared with a reference transcript), u (unknown, intergenic transcript), x (exonic overlap with reference on the opposite strand), and o (generic exonic overlap with a reference transcript) [53]. The assembled results were merged by Cuffmerge [51] and the combined transcripts were predicted by the prediction software programs CPC (version: v0.9-r2) [54], txCdsPredict (version: v423) [55], CNCI (version: v2) [55], and Pfam (version: v35) [56] database. If at least three of the four judgment methods were consistent, the transcript was confirmed to be mRNA or lncRNA. Significantly differentially expressed genes were selected with a fold change ≥ 2.00 and adjusted *p*-value ≤ 0.001 by DEGseq [57].

### 2.5. Gene Cloning and Bioinformatics Analysis

Total RNAs were extracted by the FastPure^®^ Universal Plant Total RNA Isolation Kit (Vazyme Biotech Co., Ltd., Nanjing, China). Total RNAs were reverse transcribed using the RevertAidTM First-Strand cDNA Synthesis Kit with DNase I (Vazyme Biotech Co., Ltd., Nanjing, China). Genes were amplified and cloned into the pClone007 Blunt Simple vector (Tsingke Biotechnology Co., Ltd., Beijing, China) from apricot cultivars ‘JTY’ and ‘X15’. The cis-acting elements of the *PaPSY* promoter in two apricot cultivars were predicted by PlantCARE (https://bioinformatics.psb.ugent.be/webtools/plantcare/html/ (accessed on 21 February 2023)). *PaPSY* and *PaPDS* sequence alignments were performed through the GeneDoc program and the domains were predicted by CDD (https://www.ncbi.nlm.nih.gov/Structure/bwrpsb/bwrpsb.cgi (accessed on 22 February 2023)).

### 2.6. Construction of Vectors and Transient Expression in Apricot Fruit

The full-length CDS sequences of the target genes (*PaPSY* and *PaPDS*) were amplified from apricots ‘JTY’ and ‘X15’ and constructed into the plant overexpression vector pBWA(V)HS and virus-induced gene silencing vector, respectively, and transformed into the Agrobacterium GV3101. Agrobacterium infiltration was prepared with the previously described method [58], with minor modifications. Briefly, Agrobacterium GV3101 cultured in LB at 28 °C was treated with 10 mM MES (pH 5.6), 150 μM acetosyringone, and 10 mM MgCl_2_ osmotic buffer resuspended to an OD600 = 0.8. Agrobacterium suspensions with both vectors and empty vectors (as controls) were injected into the whole apricot fruit with a 1 mL sterile syringe prior to the transcoloration stage. At least 15 fruit specimens were selected for each replicate (three biological replicates in total). The injected fruits were harvested 6–7 days after injection and then stored at −80 °C before use. The primers used for plasmid construction are listed in Appendix A.

### 2.7. GUS Staining and Activity Analysis

The *PaPSY* promoter sequence (2000 bp upstream of ATG) was amplified using ‘JTY’ and ‘X15’ as templates and connected to the pBI121 vector and transformed into Agrobacterium GV3101. Positive controls were tobacco leaves infested with Agrobacterium transfected with the pBI121 empty vector and negative controls were tobacco leaves infested with the GV3101 strain not transfected with any vector. Agrobacterium GV3101 cultured in YEP at 28 °C was treated with 10 mM MES, 150 μM AS, and 10 mM MgCl2 osmotic buffer resuspended to OD600 = 0.6 at room temperature for 3 h. This was injected into five-leaf tobacco (*Nicotiana tabacum*) leaves growing at 25 °C for 3 days. According to the previous method [59], the leaves were soaked in GUS dyeing solution overnight at 37 °C. Then, the reaction solution was removed and the leaves were decolorized with 75% ethanol until those of the control group were colorless, observed, and photographed. The primers used for plasmid construction are listed in Appendix A.

### 2.8. Validation of DEGs by qRT-PCR

A Roche LightCyler 480 instrument using 2 × SYBR Green qPCR Premix (Universal) (CodonX Biotechnology, Beijing, China) was performed in quadruplicate and the mean value ± standard deviation was used for the qRT-PCR analysis. The actin gene [60] was used as an internal reference gene to verify qRT-PCR and the relative expression of the genes was calculated according to the 2^−ΔΔCT^ method [61]. The Pearson correlation coefficients and statistical significance between the fold change among different stages from qRT-PCR and RNA-seq were calculated using SPSS 26.0. All primers are listed in Appendix A.

### 2.9. Statistical Analysis and Plotting

In analyzing the differences in β-carotene content among the different apricot cultivars and ‘JTY’ and ‘X15’ different developmental stages, a line chart of the significance of β-carotene, determined via a Student’s *t*-test, was plotted using the GraphPad Prism8 v8.0.2 software. The website https://www.omicshare.com/tools/ (accessed on 8 October 2023) was used to draw the principal component analysis diagram of 18 samples. KEGG enrichment map was drawn using R v4.2.1. The Pearson correlation coefficients (r) and statistical significance (*p*) between RNA-seq and qRT-PCR were evaluated based on their respective mean values at each stage using SPSS v26.0.1.0 software. Unless otherwise noted, figures used to display the statistics were plotted using GraphPad Prism8 v8.0.2 software.

## 3. Results

### 3.1. β-Carotene Is the Main Component That Affects the Color Difference between Orange and White Apricots

Previous studies have shown that β-carotene was the main carotenoid that determines apricot fruit color [11]. We determined the β-carotene content in the pulp of different apricot cultivars and found an apricot cultivar ‘X15’ with low β-carotene content and pure white pulp color and selected another control cultivar ‘JTY’ for follow-up experiments (Figure 1A). Both apricot cultivars ‘JTY’ and ‘X15’ have green pulp during S1–S4. Turning colors from the S5 stage: the pulp of ‘JTY’ turns from yellow-green to orange and ‘X15’ from light green to white (Figure 1B). The contents of β-carotene, chlorophyll a, and chlorophyll b in the pulps of two apricot cultivars during S1–S9 were determined (Figure 1C–E). The results indicate that β-carotene in ‘JTY’ began to significantly accumulate during S5 and reached levels 5.4-fold those of ‘X15’. The maximum value was reached during S7 and was 39 μg/g, 60.7-fold higher than that in ‘X15’. At the beginning of S8, β-carotene content slightly decreases. However, the content of β-carotene in ‘X15’ was low during the entire developmental process (Figure 1C). Furthermore, chlorophyll a content did not significantly differ between the two cultivars during S1–S3. It was significantly higher 2.4-fold in ‘X15’ compared to ‘JTY’ during S4. Moreover, the chlorophyll a content gradually decreased (*p* = 0.000) after S3 in ‘JTY’ and was not detected after S6 in ‘X15’ (Figure 1D). The content of chlorophyll b gradually decreased (*p* = 0.000) from S1 to S6 and was not detected from S7 onwards in ‘JTY’, whereas it was not detected in ‘X15’ except during S1 (Figure 1E). Therefore, we speculated that β-carotene content might be the main reason for the difference in color between ‘JTY’ and ‘X15’, especially from S5. Therefore, we conducted transcriptome sequencing with S5 as the key stages and green fruit stage (S3) and commercial maturity stage (S8) as the control stages.

### 3.2. The Transcriptional Differences between the Two Apricots Concentrated in Carotenoid Metabolism during Fruit Development

To explore the transcriptional regulation of apricot pulp color development, a total of 18 samples from the two apricot cultivars ‘JTY’ and ‘X15’ during three critical developmental periods were sequenced using RNA-seq. The Zebra sequencing platform was used to sequence 18 long RNA libraries and short reads were compared to the ribosome database using SOAPnuke (https://github.com/BGI-flexlab/SOAPnuke (accessed on 11 October 2021)). A total of 2,297,447,460 original reads were obtained after removing the reads compared with the ribosomes. Approximately 98.87% of clean reads containing adapter, excessive N, or a large number of low-quality bases was retained for subsequent analysis. The percentages of Q20 and Q30 bases were 96.80% and 92.08%, respectively, and the GC content ranged from 47.33% to 49.25%. Clean reads were compared to the reference genome GDR_Prunus_sibirica_F106 using HISAT (http://www.ccb.jhu.edu/software/hisat (accessed on 11 March 2021)). Clean reads sequenced from 18 libraries had alignment rates of 72.59–80.97% and a chain-specific ratio of 92.52–97.6% (Appendix A). The coding ability of the new transcripts was predicted by using the prediction software CPC (version: v0.9-r2), txcdspredict (version: v423), and CNC (version: v2) and the Pfam (version: v35) databases and a total of 11,557 coding transcripts and 8799 non-coding transcripts were predicted (Appendix A).

To investigate the overall difference in the transcriptome dynamics during fruit development between ‘JTY’ and ‘F43’, principal component analysis (PCA) based on Pearson’s correlation coefficient was performed using the 19,483 differentially expressed genes. The results showed that the transcriptome expression data of the two apricot cultivars at the three developmental stages were roughly divided into two groups, that is, the three stages of ‘X15’ were divided into one group and the three stages of ‘JTY’ were divided into another group (Figure 2A). In addition, correlation analysis of 18 samples showed that there were strong correlations among three biological repeats (Appendix A). The proportions of genes distributed at the four expression levels were relatively similar in all stages between the two cultivars and approximately 5.8–6.2% of genes showed very high expression levels (FPKM > 50) in different samples (Figure 2B). To investigate the details of transcriptional differences, we identified the significant differentially expressed genes (DEGs) for ‘JTY’ and ‘X15’ at each developmental stage. On the whole, 4028–6199 and 4332–6850 DEGs were significantly (fold change ≥ 2 and *p*-value ≤ 0.001) upregulated or downregulated at three comparisons of the corresponding stage in ‘JTY’ as compared with ‘X15’ (Figure 2C). We selected FDR ≤ 0.01 as the screening condition for significant enrichment and performed KEGG enrichment analysis on the differentially expressed transcripts of the three combinations. Interestingly, the carotenoid biosynthesis pathways that we focused on appeared in the top 20 pathways (Figure 2D) and next, we studied the expression of genes in this pathway during three developmental periods. To establish the authenticity of the RNA-seq data, 16 structural genes involved in the carotenoid metabolic pathways (*PSY*, *PDS*, *ZISO*, *ZDS*, *CRTISO*, *LCYB*, *CrtZ*, *ZEP*, *VDE*, *NCED*, *CCS1*, *CYP707A*, *ABA2*, *LCYE*, *LUT1*, and *LUT5*) were subjected to qRT-PCR analysis (Appendix A). The results show that the expression trends of genes were highly correlated with the RNA-seq results and were consistent (*p* < 0.05 and r > 0.8). These results verify the persuasive and mathematical significance of the transcriptome data in this study.

### 3.3. Expression Profiling of Carotenoid Biosynthesis Genes

According to KEGG enrichment results and previous studies [11], we mainly focused on the expressions of DEGs associated with carotenoid biosynthesis pathways during apricot pulp development (Figure 3). In the β-carotene synthesis pathway, the expression levels of *PaPSY* (FaF106GO300012468.01) and *PaPDS* (FaF106GO100002035.01), key genes involved in β-carotene synthesis, were higher in orange apricot ‘JTY’ than in white apricot ‘X15’ during S5. In addition, the expressions of other genes in the synthesis pathway (e.g., *ZISO*, *ZDS*, and *CRTISO*) did not significantly differ in the three key stages. Based on the β-carotene content measured during different stages, we speculated that the color transition stage (S5) was the main period of color difference between orange and white apricots. Conversely, there were no significant differences in the genes of the β-carotene degradation pathway during S5. The color differences between orange and white apricots may be due to an inhibited β-carotene synthesis pathway in white apricots.

### 3.4. PaPDS Positively Regulates β-Carotene Accumulation in Apricots

The β-carotene contents of orange apricot ‘JTY’ and white apricot ‘X15’ during different periods were measured. β-carotene levels remained low throughout the development of ‘X15’ (Figure 1C). We speculated that β-carotene might not be synthesized in ‘X15’. Therefore, we cloned all the genes in the β-carotene synthesis pathway using ‘JTY’ and ‘X15’ as templates, respectively, and found that *PaPDS* had a 5 bp deletion in exon 11 of ‘X15’ (Figure 4A). According to CDD domain prediction, the mutation was found in the *PaPDS* phytoene-desaturase functional domain, resulting in the termination of translation of this gene in ‘X15’ (Figure 4B). Thus, two overexpression vectors, pBWA-X15-PaPDS and pBWA-JTY-PaPDS, were constructed and the Agrobacterium suspension was injected into apricot pulp, with the empty pBWA vectors as controls (Figure 5A). Compared with pBWA-X15-PaPDS and pBWA, more β-carotene was accumulated in apricot fruits injected with Agrobacterium suspension containing pBWA-JTY-PaPDS. In addition, the upstream metabolites of β-carotene, phytoene, phytofluene, and lycopene accumulated more in apricot fruits injected with pBWA-JTY-PaPDS (Figure 5B). Moreover, the trends of the relative expression levels of *PaPDS* were similar to the trends of β-carotene accumulation in the injected apricot fruit (Figure 5C).

The suppression viral vector TRV1 + TRV2: PaPDS was subsequently used for transient infection of apricot pulp, with an empty TRV1 + TRV2 vector used as a control (Figure 5D). The results indicate that TRV1 + TRV2: PaPDS inhibited β-carotene and the upstream metabolites of β-carotene accumulation in apricot fruit (Figure 5E). Simultaneously, compared with the control group, *PaPDS* expression in apricot fruits injected with TRV1 + TRV2: PaPDS Agrobacterium buffer was inhibited (Figure 5F). These results indicate that the color difference between orange and white apricots may be due to the 5 bp deletion of the *PaPDS* gene in white apricot, which results in no β-carotene accumulation and thus the appearance of white apricot.

### 3.5. PaPSY May Be Regulated by Transcription Factors or lncRNA to Regulate β-Carotene Metabolism in Apricot Fruit

We cloned the *PaPSY* promoter sequence (2000 bp upstream of ATG) using ‘JTY’ and ‘X15’ as templates. Promoter cis-acting element analysis revealed quantitative differences between cha-CAM1a, MBSI, Myb, MYB, and Unnamed4 elements in the two cultivars (Figure 6A). Therefore, we constructed the two promoter sequences into the pBI121 vector, transformed Agrobacterium, and transiently transformed tobacco leaves. We found that the positive control (ck+) and both PaPSY-promoter-transgenic tobacco leaves were stained blue, except for the negative control (ck-), which was not stained, (Figure 6B). Interestingly, the PaPSY-promoter-transgenic leaves of ‘JTY’ were stained darker compared to those of ‘X15’. This indicates that the *PaPSY* promoter activity of ‘JTY’ was higher than that of ‘X15’, resulting in high expression of *PaPSY* in ‘JTY’.

Furthermore, previous studies have found that *ERF* [8], *NAC* [62], and *MYB* [2] could interact with the *PSY* promoter to regulate its expression. Therefore, we screened the transcriptome data based on the *PaPSY* expression level to identify *ERF*, *NAC*, and *MYB* transcription factors that have the same or opposite expression trend (Figure 6C). These were *PaF106G0500019206.01 (ERF1)*, *PaF106G0100000461.01 (ERF2)*, *PaF106G0200010438.01 (ERF3)*, *MTCONS_00026906 (ERF4)*, *PaF106G0800030484.01 (NAC)*, and *PaF106G0300014123.01 (MYB)*. We transiently infiltrated apricot fruit pulp with the suppressor virus vector TRV1 + TRV2: PaPSY using the empty TRV1 + TRV2 vector as a control (Figure 6D). The results show that TRV1 + TRV2: PaPSY inhibited the accumulation of β-carotene and its upstream metabolites in apricot fruits (Figure 6E). Conversely, the expression of *PaPSY* in apricot fruits was suppressed by injection of TRV1 + TRV2: PaPSY Agrobacterium buffer as compared with the control (Figure 6F). Furthermore, we focused on lncRNA targeting carotenoid biosynthesis genes and found a targeted gene pair in the β-carotene synthesis pathway, the lncRNA *LTCONS_00032302*, which was located on the antisense chain of *PaPSY*, with a total length of 550 nt, antisense regulated its putative target expression level (Figure 7A). The expression trend of *LTCONS_00032302* was consistent with that of *PaPSY* (Figure 7B,C). These results suggest that color differences between orange and white apricots may also be caused by transcription factors *ERF*, *NAC*, *MYB*, or lncRNA *LTCONS_00032302*. These transcription factors, or long non-coding RNA, regulate the expression of *PaPSY*, which leads to differences in β-carotene content.

## 4. Discussion

Carotenoids are a group of naturally occurring pigments in nature that contain a diversity of numbers and positions of carbon-dibonds that allow them to absorb different wavelengths of visible light, resulting in the visual appearance of red, orange, yellow, and green colors. The rules governing their accumulation in fruits have been well studied in peach, kiwifruit, apple, loquat, banana, and other fruit trees [2,8,63,64,65]. However, the reason for the apricot pulp color difference and the functional verification of related genes have not been reported in detail. The aim of this study was to elucidate the main mechanisms underlying the color differences between orange and white apricots, which could provide a basis for breeding improved apricot cultivars. In this study, β-carotene accumulation in orange apricot ‘JTY’ and white apricot ‘X15’ during different developmental stages were analyzed. The functions of key genes in β-carotene synthesis and the possible mechanisms of transcription factors and lncRNA regulating β-carotene accumulation were discussed. In orange apricot ‘JTY’, β-carotene content did not change during the green stages but rapidly increased at the beginning of color transformation. This highlights that the period of color transition is critical for pulp color formation in apricots, which is consistent with the results of previous studies [23]. Many other plant species also have high levels of β-carotene during color transformation, such as pepper [66], peach palm [67], carrot [9] and kiwifruits [3].

It is important to identify the key genes of β-carotene synthesis for increasing β-carotene content. We found that *PaPSY* and *PaPDS* were highly expressed in the turning stage of orange apricot ‘JTY’ by using gene expression profile analysis. The function was verified by instantaneous injection of virus-induced gene silencing and pBWA vector buffer into apricot fruit. This finding is inconsistent with previous findings. For instance, Jiang et al. [42] found that the high expression of the gene *NCED* was the main reason for the low β-carotenoid content in white apricot cultivars. Zhou et al. [23] found that *NCED* and *CCD4* were the key genes responsible for the significant difference in total carotenoid content. García et al. [44] revealed that *CCD4* was the main candidate gene for the white color of apricots. The above studies indicated that the white color of apricots were caused by different expressions of genes involved in the carotenoid degradation pathway. Nevertheless, our research found that low expression of the genes *PaPSY* and *PaPDS* (involved in carotenoid synthesis) resulted in reduced β-carotene content in white apricots. This is consistent with the study [68] that reported that tomatoes with a loss of function of the *PSY1* enzyme underwent accelerated fruit pigmentation. Furthermore, it was also found that the *MdPSY2-1* gene was directly bound and transcriptionally activated by *MdAP2-34* in apple fruit and that overexpression of *MdPSY2-1* increases phytoene and β-carotene contents in apple calli [8]. Studies on loquat [64] and carrot [69] also found that *PSY* was the main gene that determines β-carotene content. Meanwhile, in Eucommia ulmoides [70], leaf whitening could be induced by injecting the *EuPDS* silencing vector into Nicotiana benthamiana. Likewise, Zhao et al. [71] performed map-based cloning of *PDS3* and found that *PDS3* had a C-to-T substitution in the coding region, which prematurely terminated translation and resulted in yellowish-white petals in Brassica napus. Our study demonstrated a 5 bp deletion in exon 11 of *PaPDS* in ‘X15’. The truncated *PaPDS* interrupted the β-carotene biosynthesis pathway in apricot pulp, resulting in decreased β-carotene content and a white phenotype. We are developing corresponding molecular markers for this variation to provide a molecular basis for the early selection of apricot pulp color traits. The germplasm resources of pure white pulp apricot ‘X15’ screened by us can also provide a reference for breeding white apricot.

Studies on the regulatory mechanisms of β-carotene synthesis in apricots will be necessary to achieve a comprehensive understanding of the fruit’s coloring. Studies on β-carotene metabolism have generally focused on biosynthesis genes to increase the production of target metabolites. TFs play a key role in inducing the biosynthesis of specific metabolites [8,62]. In our study, we found that transient silencing of *PaPSY* could change the color of apricot fruits. Thus, we cloned the CDS sequence of the *PaPSY* gene in ‘JTY’ and ‘X15’ and found that the structures were completely consistent. We speculated that TFs might regulate the expression of *PaPSY*. The *PaPSY* promoters were predicted and the numbers of cha-CAM1a, MBSI, and Myb elements differed between the two cultivars. The expression levels of *PaPSY*, *PaERF*, *PaNAC*, and *PaMYB* were highly correlated with carotenoid synthesis. Moreover, the *PaPSY* promoter activity of ‘JTY’ was higher than that of ‘X15’. Previous studies have also reported that in apples, *MdAP2-34* can bind the ACCGAC motif in the *MdPSY2-1* promoter [8]. In Arabidopsis, *AtRAP2.2* binds to ATCTA, a cis-acting element on the *AtPSY* promoter, promoting the expression of this gene [72]. The regulatory mechanism of transcription factors on *PaPSY* remains to be further explored. Furthermore, LncRNA-mediated regulation has previously been associated with carotenoid biosynthesis. In tomatoes, the carotenoid biosynthesis gene *LCYB* was significantly upregulated in *lncRNA1459* functionally deficient mutants compared with wild-type fruits [73]. In sea buckthorn, the expressions of three lncRNAs (*XLOC_267510*, *XLOC_338163*, and *XLOC_169881*) significantly differed between yellow and red fruit and were involved in the carotenoid biosynthesis pathway [74]. In our study, we propose a regulatory framework in which lncRNAs participate in carotenoid biosynthesis by positively regulating their cis-target mRNA. Interestingly, *LTCONS_00032302*, located on the *PaPSY* antisense chain, positively regulates the expression of *PaPSY*, which may promote the accumulation of β-carotene. Further experiments are needed to verify the regulatory mechanism of *LTCONS_00032302* on *PaPSY*.

In summary, we established a simple model that explains the mechanism by which *PaPSY* and *PaPDS* regulate the color of apricot pulp (orange vs. white) (Figure 8). That is, the absence of 5 bp in the *PaPDS* coding region in white apricots prevents β-carotene synthesis and results in white pulp. In addition, it is possible that transcription factors or lncRNA regulate the expression of *PaPSY*, which affects β-carotene accumulation in orange and white apricots. These results provide us with a new and comprehensive landscape of carotenoid regulation in apricots and can be further used in ongoing molecular breeding programs.

## 5. Conclusions

The content of β-carotene affects the color of apricot pulp. In this study, we screened the β-carotene synthesis pathway genes *PaPSY* and *PaPDS.* There was a 5 bp deletion of exon 11 of *PaPDS* in ‘X15’; gene overexpression and virus-induced silencing analysis showed that truncated *PaPDS* disrupted the β-carotene biosynthesis pathway in apricot pulp, resulting in decreased β-carotene content and white pulp phenotype. Furthermore, virus-induced silencing analysis showed that *PaPSY* was also a key gene in β-carotene biosynthesis. It provides an important point of view for analyzing the root cause of the formation of apricot pulp color traits and a theoretical reference for the cultivation of new apricot cultivars in production applications.

## Figures and Tables

**Figure 1 foods-13-00300-f001:**
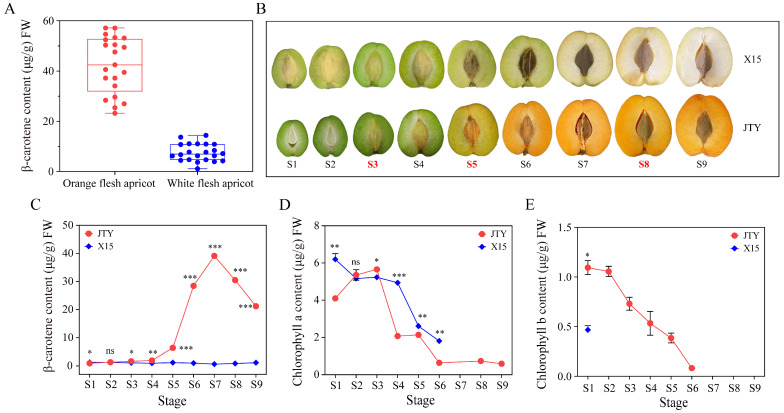
Phenotypic β-carotene and chlorophyll a and b content across different developmental stages in ‘X15’ and ‘JTY’. (**A**) β-carotene content of different orange and white pulp apricot cultivars. (**B**) Phenotypic changes of JTY’ and ‘X15’ at different developmental stages. The contents of β-carotene (**C**), chlorophyll a (**D**), and chlorophyll b (**E**) at different developmental stages. Note: * *p* < 0.05, ** *p* < 0.01, *** *p* < 0.001, ns: not significant.

**Figure 2 foods-13-00300-f002:**
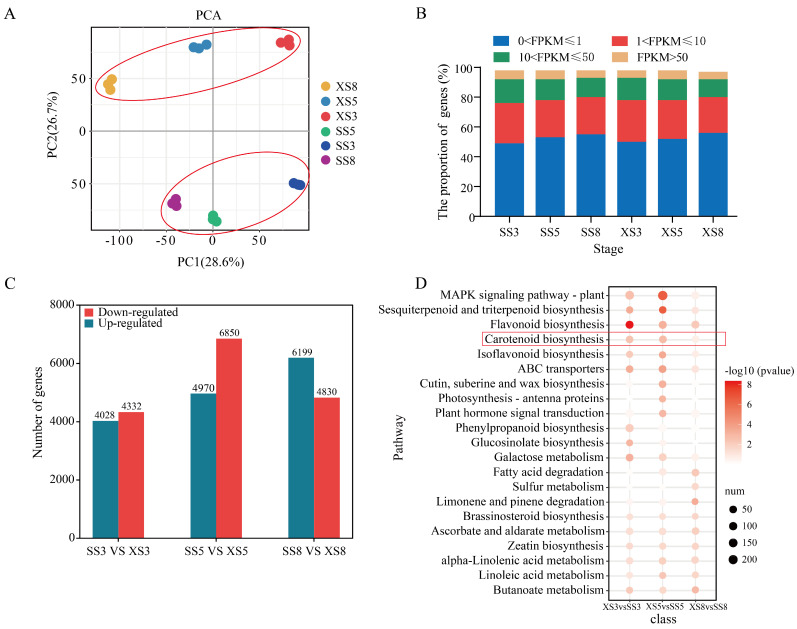
Preliminary analysis of transcriptome data in ‘X15’ and ‘JTY’. (**A**) Principal component analysis of 18 samples based on FPKM. (**B**) The proportion of expressed genes at four different expression levels in ‘JTY’ (SS3, SS5, and SS8) and ‘X15’ (XS3, XS5, and XS8). (**C**) The numbers of upregulated and downregulated genes at three stages of fruit development in ‘JTY’ as compared with ‘X15’. (**D**) KEGG enrichment of DEGs. The X-axis indicates comparative combinations, the Y-axis indicates enriched KEGG items, size indicates number of identified genes in the background, and color indicates significance level.

**Figure 3 foods-13-00300-f003:**
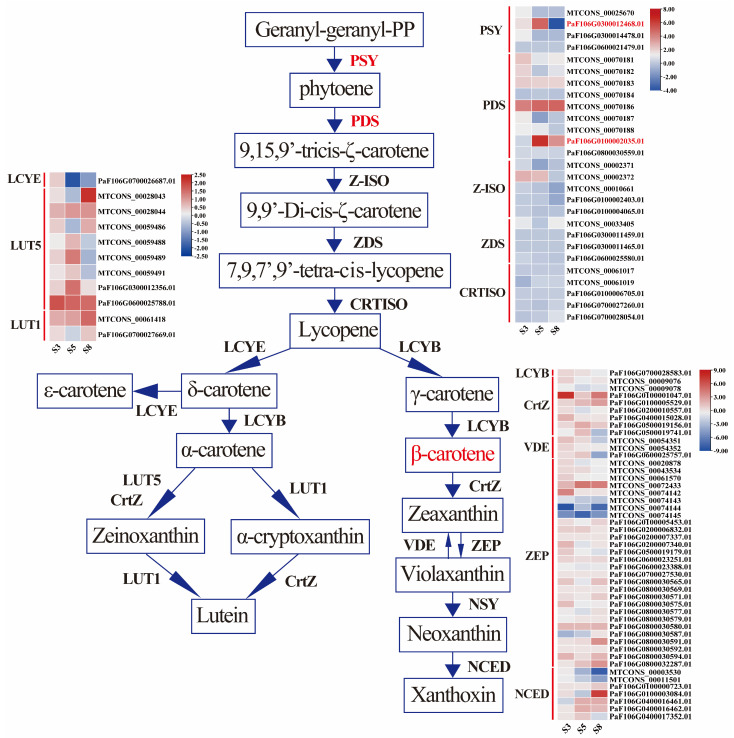
Schematic diagram and heat maps of differentially expressed genes (DEGs) involved in carotenoid metabolic pathways during different developmental stages in ‘X15’ and ‘JTY’. DEGs linked to enzymes and their log2FC values are presented in the heat maps. Redder and bluer colors represent higher and lower log2FC values, respectively. PSY, 15-*cis*-phytoene synthase; PDS, 15-*cis*-phytoene desaturase; ZISO, zeta-carotene isomerase; ZDS, zeta-carotene desaturase; CRTISO, prolycopene isomerase; LCYB, lycopene beta-cyclase; CrtZ, beta-carotene 3-hydroxylase; ZEP, zeaxanthin epoxidase; VDE, violaxanthin de-epoxidase; NSY, neoxanthin synthase; NCED, 9-*cis*-epoxycarotenoid dioxygenase; LCYE, lycopene epsilon-cyclase; LUT5, beta-ring hydroxylase; LUT1, carotenoid epsilon hydroxylase.

**Figure 4 foods-13-00300-f004:**
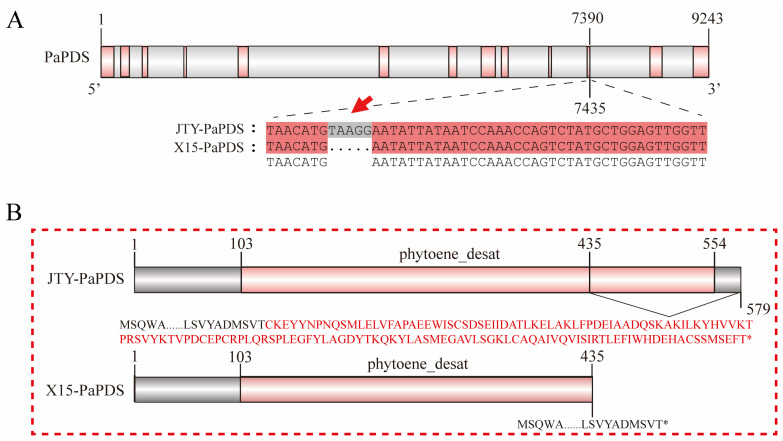
Structure analysis of the PaPDS gene. (**A**) Schematic diagram of PaPDS base mutation sites in ‘X15’ and ‘JTY’, the red arrow indicates a 5 bp mutation. Note: red indicates the CDS of the PaPDS gene, and gray indicates the introns. (**B**) Schematic diagram of the PaPDS amino acid domain in ‘X15’ and ‘JTY’. Note: red indicates the functional domain of PaPDS, * indicates termination of translation.

**Figure 5 foods-13-00300-f005:**
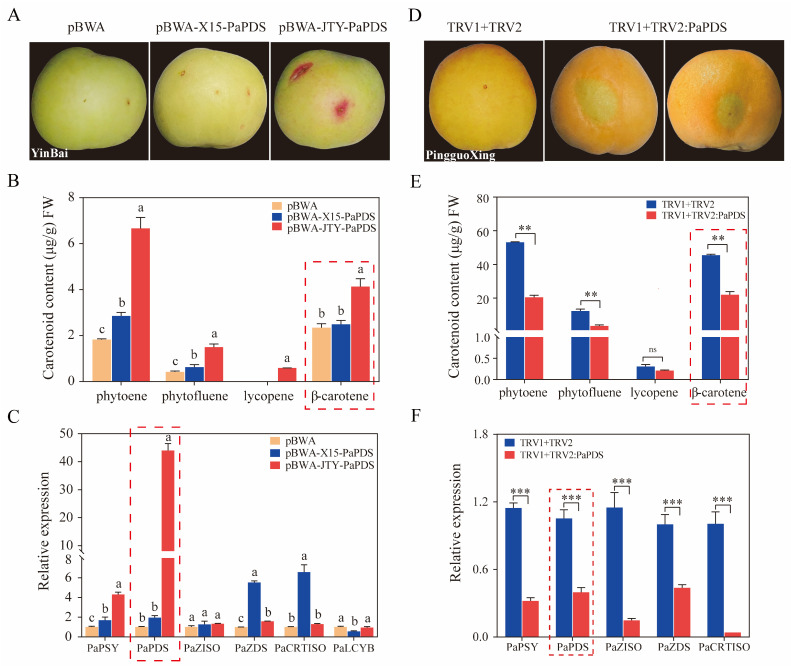
PaPDS positively regulates β-carotene accumulation in apricots. (**A**) Coloration of ‘Yinbai’ injected with PaPDS overexpression Agrobacterium plasmid mixture (pBWA-X15-PaPDS, pBWA-JTY-PaPDS). An empty pBWA vector was used as a control. (**B**) Determination of carotenoid content in overexpression experiments. (**C**) Relative expression levels of β-carotene synthesis pathway genes in overexpression experiments. (**D**) Coloration of ‘Pingguoxing’ injected with PaPDS silencing Agrobacterium plasmid mixture (TRV1 + TRV2: PaPDS). An empty TRV1 + TRV2 vector was used as a control. (**E**) Determination of carotenoid content in silencing experiments. (**F**) Relative expression levels of β-carotene synthesis pathway genes in silencing experiments. The different letters represent significant differences (LSD test, *p* < 0.05). The asterisks indicate significant differences (** *p* < 0.01, *** *p* < 0.001, ns: not significant) based on the Student’s *t*-test.

**Figure 6 foods-13-00300-f006:**
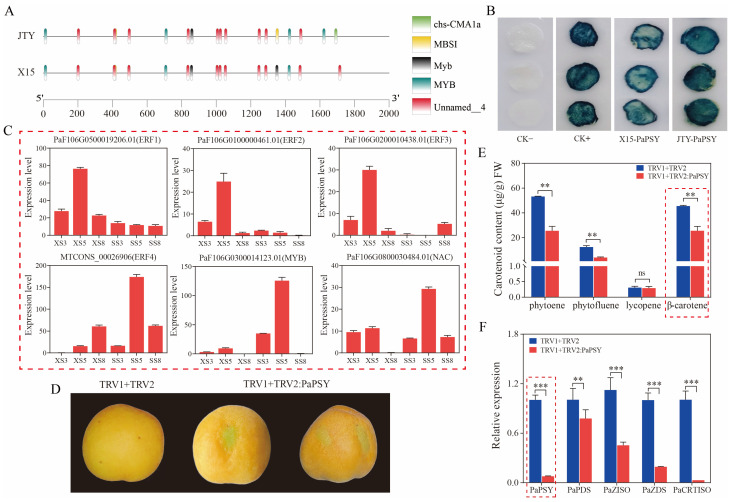
PaPSY may be regulated by transcription factors to influence β-carotene metabolism in apricots. (**A**) Schematic diagram of cis-acting elements of the PaPSY promoter in ‘X15’ and ‘JTY’. (**B**) GUS staining for transient expression of the PaPSY promoter of ‘X15’ and ‘JTY’ in tobacco leaves. (**C**) Expression levels of possible transcription factors regulating PaPSY. (**D**) Coloration of ‘Pingguoxing’ injected with PaPSY silencing Agrobacterium plasmid mixture (TRV1 + TRV2: PaPSY). An empty TRV1 + TRV2 vector was used as a control. (**E**) Determination of carotenoid content in silencing experiments. (**F**) Relative expression levels of β-carotene synthesis pathway genes in silencing experiments. The different letters represent significant differences (LSD test, *p* < 0.05). The asterisks indicate significant differences (** *p* < 0.01, *** *p* < 0.001, ns: not significant) based on the Student’s *t*-test.

**Figure 7 foods-13-00300-f007:**
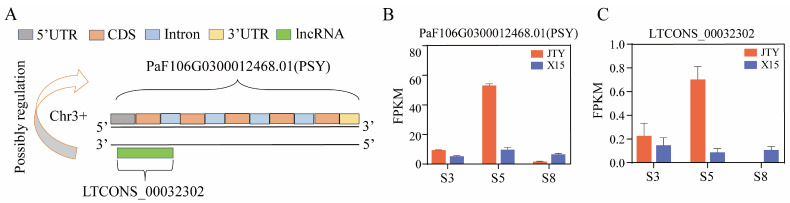
PaPSY may be regulated by lncRNA to influence β-carotene metabolism in apricots. (**A**) Pattern diagram of lncRNA regulation of genes related to the carotene metabolism pathway. (**B**) Expression of PaPSY at three growth and developmental stages in ‘JTY’ and ‘X15’. (**C**) Expression of LTCONS_00032302 at three growth and developmental periods in ‘JTY’ and ‘X15’.

**Figure 8 foods-13-00300-f008:**
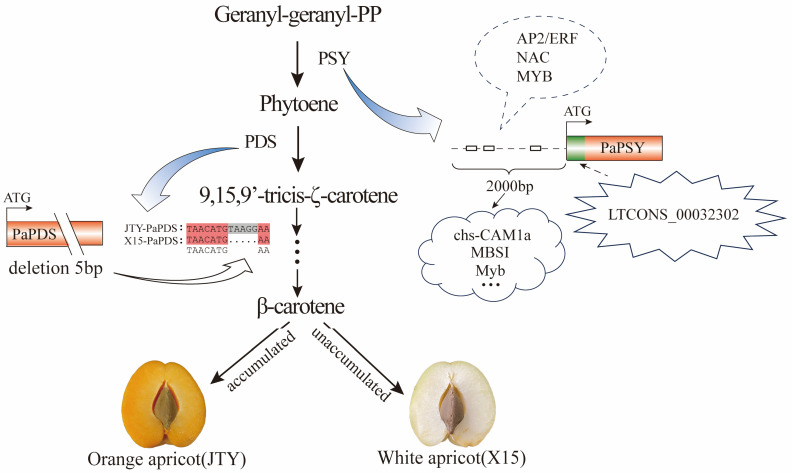
Working model for PaPSY and PaPDS function in the regulation of ’JTY’ and ‘X15’. Dashed balloon indicates that ERF, NAC, and MYB may bind to cha-CAM1a, MBSI, Myb, and other cis-acting elements on the PaPSY promoter, thereby affecting the expression and function of PaPSY. Dotted arrows indicate that LTCONS_00032302 targets PaPSY, which may affect the expression and function of PaPSY.

## Data Availability

The raw transcriptome sequencing reads in this research were uploaded to the NCBI SRA database (PRJNA1000057).

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
