# Peer review of "15-cis-Phytoene Desaturase and 15-cis-Phytoene Synthase Can Catalyze the Synthesis of β-Carotene and Influence the Color of Apricot Pulp"

_foods, 2024, doi:10.3390/foods13020300_

Round 1

Reviewer 1 Report

Comments and Suggestions for Authors

Dear Editor,

I have carefully reviewed the manuscript entitled "PaPSY and PaPDS can catalyze the synthesis of β-carotene and influence the color of apricot flesh." I regret to inform you that, in its current state, the manuscript is not suitable for publication due to several concerns outlined below.

Lack of Novelty and Limited Contribution to Advancements:

The manuscript appears to lack novelty, and it does not significantly contribute to the existing body of knowledge in the field. The study does not seem to introduce new methodologies, concepts, or findings that would advance the understanding of the subject matter.

Insufficient Duration of Study:

The one-year duration of the study raises concerns about the reliability and generalizability of the results. Fruit color and gene expression can be influenced by various factors, including environmental conditions, over an extended period. A more comprehensive study, conducted over multiple years, would provide a more robust foundation for drawing conclusions and establishing the validity of the reported results.

Limited Applicability Beyond Experimental Region:

The manuscript fails to address the broader implications of the study, as it is described as a local work with data valid only for the experimental region. To enhance the significance of the findings, the authors should discuss how the results could be extrapolated or applied to similar regions or conditions. This could involve a more thorough examination of the environmental variables and a discussion of the potential impact on different geographical locations.

Sincerely,

Reviewer 2 Report

Comments and Suggestions for Authors

Thank you for submitting the manuscript "PaPSY and PaPDS can catalyze the synthesis of β-carotene and influence the color of apricot flesh" to Foods.

 The manuscript describes specific genes for carotenoid synthesis in apricot. Overall, the manuscript is well written but needs to be revised regarding technical issues as it uses some terminologies and/or the way it is written does not correspond to scientific writing.

 Additionally, I have a few points:

 Line#3: did you mean fresh apricot? In every text it is necessary to standardize the way of defining the object of study.

 Line#33: this statement makes no sense.

 Line#41: rich nutrition is a very superficial expression that conveys little scientific information.

 Line#66: I believe you meant here "the synthesis of lycopene is catalyzed by..."

 Were the results described as "increased" or "decreased" in the text based on statistical analysis? If yes, please add the actual p in parentheses after this statement.

 Line#108: how was the S1 stadium defined? Why 49 days?

 Line#236: why were only these stages used for the transcription study? An explanation must be added.

 In general, the figure captions are too large. It would be possible to reduce by considering all relevant information.

 Line#433: cite previous studies

 Check scientific names throughout the manuscript.

 - results and discussion as separate items is not common in this type of manuscript and I believe it hinders the discussion of results somewhat. My suggestion is that these items are just one item.

Comments on the Quality of English Language

Moderate editing of English language required. 

Reviewer 3 Report

Comments and Suggestions for Authors

Dear Editor Natalija Knežević

Manuscript ID: foods-2798747

Manuscript title: PaPSY and PaPDS can catalyze the synthesis of β-carotene and influence the color of apricot flesh

I completed review of the manuscript with title “PaPSY and PaPDS can catalyze the synthesis of β-carotene and influence the color of apricot flesh”. The research is well written, and contains detailed information and original findings. I think it is a very valuable research. There are findings that will interest the reader. Thus, the manuscript could be acceptable for publication in FOODS with minor corrections. But research needs to be improved. In particular, the research question and hypothesis of the study are missing and need to be added. My suggestions were shown on PDF file

With my best regards

Round 2

Reviewer 1 Report

Comments and Suggestions for Authors

Accept in present form